# RNA-Seq Analysis of Cisplatin and the Monofunctional Platinum(II) Complex, Phenanthriplatin, in A549 Non-Small Cell Lung Cancer and IMR90 Lung Fibroblast Cell Lines

**DOI:** 10.3390/cells9122637

**Published:** 2020-12-08

**Authors:** Jerry D. Monroe, Satya A. Moolani, Elvin N. Irihamye, Joshua S. Speed, Yann Gibert, Michael E. Smith

**Affiliations:** 1Department of Cell and Molecular Biology, University of Mississippi Medical Center, Jackson, MS 39216, USA; jmonroe1@umc.edu (J.D.M.); ygibert@umc.edu (Y.G.); 2Department of Biology, Western Kentucky University, Bowling Green, KY 42101-1080, USA; satya.moolani594@topper.wku.edu (S.A.M.); elvin.irihamye110@topper.wku.edu (E.N.I.); 3Program in Cognitive Science, Case Western Reserve University, Cleveland, OH 44106-7063, USA; 4Program in Neuroscience, Indiana University Bloomington, Bloomington, IN 47405-2204, USA; 5Department of Physiology and Biophysics, University of Mississippi Medical Center, Jackson, MS 39216, USA; jspeed@umc.edu

**Keywords:** next generation sequencing, cancer, monofunctional platinum(II) complex, pathway analysis, cisplatin

## Abstract

Phenanthriplatin is a new monofunctional platinum(II) complex that binds only one strand of DNA and acts by blocking gene transcription, but its effect on gene regulation has not been characterized relative to the traditional platinum-based complex, cisplatin. A549 non-small cell lung cancer and IMR90 lung fibroblast cells were treated with cisplatin, phenanthriplatin, or a control and then their RNA transcripts were subjected to next generation sequencing analysis. DESeq2 and CuffDiff2 were used to identify up- and downregulated genes and Gene Ontology and Kyoto Encyclopedia of Genes and Genomes databases were used to identify pathways and functions. We found that phenanthriplatin may regulate the genes GPRC5a, TFF1, and TNFRSF10D, which act through p53 to control apoptosis, differently or to a greater extent than cisplatin, and that it, unlike cisplatin, could upregulate ATP5MD, a gene which signals through the Wnt/β catenin pathway. Furthermore, phenanthriplatin caused unique or enhanced effects compared to cisplatin on genes regulating the cytoskeleton, cell migration, and proliferation, e.g., AGAP1, DIAPH2, GDF15, and THSD1 (*p* < 0.05; *q* < 0.05). Phenanthriplatin may modulate some oncogenes differently than cisplatin potentially leading to improved clinical outcome, but this monofunctional complex should be carefully matched with cancer gene data to be successfully applied in chemotherapy.

## 1. Introduction

The platinum(II) chemotherapy drug, cis-diamminedichloroplatinum(II) (cisplatin) is a bifunctional complex with two chloride leaving ligands which typically forms intrastrand crosslinks with DNA resulting in altered gene transcription [1,2] (Figure 1). Studies employing microarrays and next generation sequencing (NGS) show that transcriptional changes induced by cisplatin include effects on genes associated with apoptosis, drug resistance, metabolism, cell proliferation, cell adhesion, stress response, cell cycle control, and DNA repair [3,4,5,6,7,8,9]. Unfortunately, cisplatin treatment can also cause apoptosis in non-cancerous cells leading to severe side-effects by damaging renal, auditory and nervous system tissue and also activate chemotherapy resistance mechanisms that prevent its anti-cancer activity [1,10,11].

Several monofunctional platinum(II) complexes have recently been investigated as anti-cancer drug candidates. The monofunctional complex, cis-[Pt(NH_3_)_2_Cl (phenanthridine)]^+^ (phenanthriplatin, Figure 1) has similar or even superior anticancer effect than cisplatin in various cancer cell lines [12,13]. Unlike cisplatin, monofunctional complexes have only one chloride leaving ligand and bind to a single strand of DNA without causing DNA to bend [12,14]. Treatment with phenanthriplatin can alter RNA transcription in exogenous reporters expressed in cancer cells and change RNA polymerase function [12,15]. Furthermore, a polymerase halting assay has shown that phenanthriplatin has a distinct DNA residue binding profile from that of cisplatin with the monofunctional complex able to bind adenine residues to a greater extent than cisplatin but that both preferentially bind to guanine residues [16]. As this monofunctional complex may have a unique mode of action to block transcription and could potentially target different genes than cisplatin, it may cause increased cytotoxicity in malignant cells while producing fewer side-effects and without promoting resistance mechanisms. However, the effects of monofunctional complexes on gene expression have not yet been characterized.

This study used next generation sequencing (NGS; Figure 2) to determine whether phenanthriplatin induces a different gene expression profile than the bifunctional complex, cisplatin. An understanding of the differential effect of monofunctional vs. bifunctional complexes on gene expression will provide insight on how these two different drugs inhibit cancer growth and provide insight on differences in off-target effects. A549 non-small cell lung cancer (NSCLC) cells were selected for study as they are routinely used to evaluate platinum-based gene modulation, drug toxicity and resistance, and the non-cancer lung fibroblast cell line, IMR90, which has also been used to assess cisplatin toxicity and gene regulation [9,13,17]. We treated A549 and IMR90 cells with either phenanthriplatin, cisplatin, or a negative solvent only control. Then, total RNA was extracted from the cell lines, processed and subjected to NGS analysis. DESeq2 analysis and Cuffdiff2 was used to generate lists of the top 10 up- and downregulated genes for treatment comparisons between and within cell lines and to compare the effect of both compounds with their controls and each other. We also used the R package clusterProfiler to elucidate the top 20 enriched biological processes and pathways using Gene Ontology (GO) and Kyoto Encyclopedia of Genes and Genomes (KEGG) analytical tools. Droplet digital polymerase chain reaction (ddPCR) was used to validate the NGS results for selected up- and downregulated genes. Our data set shows that phenanthriplatin can modulate genes whose expression was not altered by cisplatin treatment in both A549 and IMR90 cells. Furthermore, the monofunctional complex can target gene expression in one cell line and not the other, suggesting that it may have different effects in NSCLC cells as opposed to normal lung fibroblast cells. Perhaps most interesting, we found that phenanthriplatin can cause a gene expression effect opposite to that of cisplatin on several genes in both cell lines, indicating that the monofunctional complex may signal through distinct cell signaling mechanisms to affect cellular function. Additionally, chemotherapeutic usage of this monofunctional complex should carefully consider the genetic characteristics of the cancer and genes related to potential side-effects in non-cancerous tissues.

## 2. Materials and Methods

### 2.1. Cell Culture and Treatment

The A549 (non-small cell lung cancer) and IMR90 (non-cancerous lung fibroblast) cell lines were obtained from ATCC (Manassas, VA, USA). A549 cells were cultured in F12K media with 10% FBS and 1% penicillin/streptomycin supplementation. IMR90 cells were cultured in Eagle’s Minimum Essential Medium (EMEM) with 10% FBS and 1% penicillin/streptomycin supplementation. 6-well dishes were seeded with 3 × 10^5^ cells per well and placed in an incubator (37 °C, 5% CO_2_) for 24 h. Sets of three wells each were treated with either a negative control (media only), or 5 µM cisplatin or phenanthriplatin (5 µM cisplatin and phenanthriplatin is a concentration which is efficiently subject to uptake in the nucleus of A549 and IMR90 cells within a 24 h interval [12,13]). Then, the dishes were placed back into the incubator for 24 h after which they were prepared for RNA isolation.

### 2.2. RNA Isolation and Library Preparation

The Qiagen RNeasy kit (Hilden, Germany) was used to isolate RNA samples from control and platinum compound-treated dishes per manufacturer’s instructions. All RNA samples were stored at −80 °C for subsequent analysis. Libraries were prepared using the NuGEN Technologies (Redwood City, CA, USA) Universal Plus mRNA-Seq kit with NuQuant (NuGEN Cat. #0508) per manufacturer’s instructions. cDNA was purified using 1.8 volumes Agencourt AMPure XP Beads (Beckman Coulter, Brea, CA, USA), eluted in 10 µl of H_2_O, and stored at −20 °C. Samples were barcoded with NuGEN Universal Plus mRNA-Seq adaptors as listed in Appendix A. Eluted libraries were collected and stored at −20 °C. The concentration of the libraries was validated using NuQuant assays. Libraries were diluted and normalized to the optimal range for Agilent Bioanalyzer (Santa Clara, CA, USA) analysis using the DNA High Sensitivity Kit (cat. no. 5067-4626, Agilent Technologies, ). The average library length was equal to 353 bp. Libraries were normalized and pooled based on molar concentration values obtained from bioanalyzer measurements.

### 2.3. Next Generation Sequencing 

The pooled library was prepared and sequenced using the Illumina (San Diego, CA, USA) MiSeq Reagent Nano Kit V2 300 cycles kit (cat. no. MS-103-1001). Library samples and PhiX control (cat. no. FC-110-3001) were denatured and diluted using the standard normalization method according to the manufacturer’s directions. 50% PhiX was spiked in and sequencing was performed using an Illumina MiSeq Nano 300 to test quantity and quality. Library and PhiX samples for sequencing were denatured and diluted according to manufacturer’s directions. Two sequencing runs were performed on an Illumina NextSeq 500 using the NextSeq 500/550 75 cycle High Output Kit v2.5 (cat. no. 20024906).

### 2.4. Droplet Digital Polymerase Chain Reaction Analysis

Next Generation Sequencing results were validated using droplet digital polymerase chain reaction (ddPCR) on project RNA samples. First, 100 or 1000 ng samples of RNA were converted to cDNA with the iScript Reverse Transcription kit (BioRad Laboratories, Hercules, CA, USA). Samples were then placed in a thermocycler (BioRad) and run according to kit instructions. Then, the ddPCR Supermix for Probes kit (BioRad) was used to attach probes to cDNA from 5 or 50 ng of total RNA. The reaction mix was then separated into nanodroplets using the Automated Droplet Generator (BioRad). Next, PCR was carried out for 40 cycles per manufacturer’s instructions. Droplets were then analyzed using the QX200 Droplet Reader (BioRad) and data was quantified and copy count calculated using QuantaSoft (BioRad) software.

### 2.5. Data Analysis

Sequenced samples were then comparatively analyzed according to the following pairwise comparisons with the second group used as the baseline: A549 phenanthriplatin versus A549 control, A549 cisplatin versus A549 control, IMR90 phenanthriplatin versus IMR90 control, IMR90 cisplatin versus IMR90 control, A549 phenanthriplatin versus A549 cisplatin, and IMR90 phenanthriplatin versus IMR90 cisplatin.

192 single-end raw sequencing files (.fastq) were downloaded from Illumina’s BaseSpace (https://basespace.illumina.com/) onto the Kentucky Biomedical Research Infrastructure Network (KBRIN) server for analysis. Eighteen samples (2 cell lines × 3 replicates × 3 treatments) were submitted for analysis across four sequencing lanes in two sequencing runs. Sequence read quality was then determined by concatenating the 192 single-end raw .fastq files across sequencing lanes for each replicate into one single-end .fastq file using the unix cat command. Eighteen files, representing six samples and three biological replicates each, were generated for each sequencing run. Quality control of raw sequence data was performed using FastQC (version 0.10.1) for each sequencing run. The FastQC results showed that the sequences were of high quality throughout (Appendix A), and no sequence trimming was necessary. Next, the concatenated sequences were directly aligned to the *Homo sapiens* hg38 reference genome assembly (hg38.fa) using STAR (version 2.6), generating alignment files in bam format. The alignment rate was above 99% for all samples; the number of raw reads successfully aligned for each of the samples is shown in Appendix A.

Differential expression analysis was performed using DESeq2 and Cuffdiff2. For DESeq2, raw counts were obtained from the STAR aligned bam format files using HTSeq version 0.10.0. The raw counts were normalized with DESeq2 using a scaling factor based on median gene expression across the samples (Anders and Huber, 2010 [18]), expressed using the relative log expression (RLE) method, and then filtered to exclude genes with fewer than 10 counts across the samples. For Cuffdiff2 analysis, Cuffnorm was used to produce FPKM (Fragments Per Kilobase Million) normalized counts. The counts were then filtered to include only genes with a minimum expression of one FPKM in three or more samples and an average expression of at least one FPKM. Also, the R package clusterProfiler was used to identify enriched Gene Ontology (GO) biological processes and Kyoto Encyclopedia of Genes and Genomes (KEGG) pathways for each set of differentially expressed genes. Volcano plots were also created for each comparison to examine the distribution of log2 fold change at different significance levels.

### 2.6. Statistical Analysis

Differential expression (DESeq2) results were analyzed using a *p* < 0.05 significance level followed by performance of a false discovery rate analysis (*q* < 0.05). Analysis of GO processes and KEGG pathways using clusterProfiler generated adjusted *p* values. Volcano plots were analyzed using *p* and *q* value significance levels of < 0.05. Digital droplet PCR data was statistically analyzed with GraphPad PRISM version 8.4.2 (La Jolla, CA, USA) using a two-way ANOVA with Dunnett’s multiple comparison test with a *p* < 0.05 significance level.

## 3. Results

DESeq2 and Cuffdiff2 analysis was performed on next generation sequencing samples to obtain differentially expressed gene (DEG) profiles in phenanthriplatin, cisplatin, and control treated A549 and IMR90 cells. First, we used derived log2 fold change values to identify the most up- and downregulated genes in A549 cells treated with phenanthriplatin compared to its control. We found that several genes were up- and downregulated by phenanthriplatin (Table 1). Cisplatin treatment also up- and downregulated genes in A549 cells versus the control treatment category (Table 2). We then used GO analysis to identify the 20 most enriched biological processes in the A549 phenanthriplatin versus A549 control comparison and found that the monofunctional complex regulated a variety of cellular processes (Figure 3A). Similarly, GO analysis of the enriched biological processes in A549 cisplatin versus control cells showed that a large variety of cellular processes were activated (Figure 3B). KEGG pathway analysis showed that both phenanthriplatin and cisplatin modulated a large set of pathways involving a variety of diseases (Figure 3C,D). As a final measure we plotted the distribution of regulated genes using a volcano plot format and found that phenanthriplatin modulated genes more highly than did cisplatin compared to controls in A549 cells (Figure 4A,B,G).

Next, in order to evaluate the effects of phenanthriplatin and cisplatin on non-cancer lung cells, we used derived log2 fold change values to identify the most up- and downregulated genes in IMR90 cells treated with either of the two complexes compared to their controls. We found that the monofunctional complex up- and downregulated several genes (Table 3). In IMR90 cells, we also found that the bifunctional complex up- and downregulated several genes (Table 4). As with A549 cells, we used GO analysis to identify the 20 most enriched biological processes in the IMR90 phenanthriplatin versus IMR90 control comparison and found that the monofunctional complex regulated a variety of cellular processes (Figure 5A). Similarly, GO analysis of the enriched biological processes in IMR90 cisplatin versus control cells identified several processes mostly overlapping those targeted by phenanthriplatin (Figure 5B). KEGG pathway analysis showed that phenanthriplatin modulated pathways involving a variety of diseases (Figure 5C) as did cisplatin (Figure 5D). When we plotted IMR90 genes using a volcano plot format, we found that as before with A549 cells, phenanthriplatin modulated genes more highly than did cisplatin compared to IMR90 control samples (Figure 4C,D).

We also performed a comparative analysis of the two platinum complexes in either cell line and found that in the A549 cell line, there were several genes that were more up- or downregulated by phenanthriplatin compared to cisplatin (Table 5), and in the IMR90 cell line, phenanthriplatin also up- and downregulated a set of genes more than cisplatin (Table 6). Next, we used GO analysis to identify the 20 most enriched biological processes in the A549 phenanthriplatin versus A549 cisplatin comparison and found that the monofunctional complex regulated a distinct set of cellular processes to a greater degree than the bifunctional complex (Figure 6A). Similarly, GO analysis of the enriched biological processes in IMR90 phenanthriplatin versus IMR90 cisplatin cells showed that the monofunctional complex effected several cellular processes differently than did cisplatin (Figure 6B). KEGG pathway analysis showed that phenanthriplatin compared to cisplatin in A549 cells modulated pathways involving a variety of diseases (Figure 6C) and that in IMR90 cells, phenanthriplatin compared to cisplatin also modulated similar pathways to those in the A549 comparison including the same nervous system diseases, but more cancer and platinum cancer resistance pathways (Figure 6D). When we plotted the A549 and IMR90 phenanthriplatin versus cisplatin comparisons using the volcano plot format, we found that gene regulation was very similar in both treatment categories (Figure 4E,F).

In order to validate the next generation sequencing data, we performed ddPCR on genes subject to the highest degree of up- and downregulation in the different treatment comparisons. We found that in A549 cells, phenanthriplatin reduced the expression of AGAP1, DIAPH2, GPRC5A, THSD4, cisplatin increased GDF15, the monofunctional complex increased TFF1 expression, phenanthriplatin decreased, but cisplatin increased THSD1 expression, while neither complex altered SAT1 expression (Figure 7). In IMR90 cells, both platinum complexes decreased AGAP1, DIAPH2, and THSD4 expression, and increased GDF15 and SAT1, and cisplatin treatment only increased GPRC5A, TFF1 and THSD1 (Figure 7). Results for the remaining ddPCR experiments are in Appendix A. Comparison of the NGS gene expression data with the ddPCR data showed that both data sets were consistent with one another (Table 1, Table 2, Table 3, Table 4, Table 5 and Table 6, Figure 7 and Appendix A).

## 4. Discussion

Monofunctional complexes which block gene transcription without inducing the DNA structural distortion caused by bifunctional complexes could modulate distinct genes or act through unique gene networks. Their unique mode-of-action could allow them to have anticancer efficacy against NSCLC cells without causing negative side-effects or developing chemotherapy resistance normally associated with traditional platinum-based chemotherapy. Cisplatin is primarily subject to cellular uptake via the copper transporter, Ctr1 [19], while phenanthriplatin has a high affinity for organic cation transporter 2, but unlike cisplatin, also efficiently targets multidrug and toxin extrusion proteins which could facilitate its efflux and reduce its efficacy [20]. These unique cell uptake and efflux modalities suggest that these complexes could have distinct kinetic and potency profiles stemming from different nuclear compartmental targeting efficiencies. In A549 cells, the monofunctional complex exhibits greater uptake of platinum in both the cytosolic and nuclear compartments than cisplatin [12]. Furthermore, the intracellular distribution of phenanthriplatin in the cytosol and nucleus of A549 and IMR90 cells is very similar, suggesting that the monofunctional complex may target the nuclear compartment with similar efficiency in NSCLC and normal lung fibroblast cells [13]. Cisplatin and phenanthriplatin preferentially target guanine residues, but the monofunctional complex has a greater binding affinity to adenine than cisplatin [16]. Studies of cisplatin and phenanthriplatin binding rates to guanine nucleotide analogs have revealed that the monofunctional complex has a half-life less than half of cisplatin [12,21]; although binding rates for phenanthriplatin to guanine nucleotide analogs in A549 and IMR90 cells may not correlate well with effects against cell viability [13]. In A549 cells, phenanthriplatin has a stronger molecular potency (IC_50_ values: 0.058 to 0.22 µM) than cisplatin (6.75 to 9.79 µM) [12,13], while in IMR90 cells both complexes have similar potencies (cisplatin, 0.53 µM; phenanthriplatin, 1.24 µM) [13]. As cisplatin and phenanthriplatin have distinct uptake, reaction kinetic, nucleotide targeting effects, and molecular potencies in NSCLC cells, we used NGS analysis on samples treated for 24 h, a time point when cisplatin strongly modulates gene expression in cancer cells [22], to identify genes that were regulated similarly by both complexes, by phenanthriplatin alone or that the monofunctional complex caused an opposite effect upon compared to its bifunctional counterpart.

We found in A549 and IMR90 cells that cisplatin and phenanthriplatin both upregulated ATF3 (activating transcription factor 3) more than any other gene (Table 1, Table 2 and Table 3). Overexpression of ATF3 is associated with apoptosis and promotes cytotoxicity in cisplatin treated A549 cells via p53 signaling [23]. Furthermore, expression of N-myc downstream regulated gene 1 (NDRG1), a gene that is regulated by hypoxia and cellular proliferation signaling, and can act to both promote and suppress some cancers, has been shown to counteract the function of ATF3 [23]. These results suggest that both cisplatin and the monofunctional complex can potentially suppress NDRG1’s action to suppress ATF3. However, our results also suggest that phenanthriplatin might act against IMR90 cells; although it is uncertain whether NDRG1’s activity in normal cells would be subject to similar hypoxia and cell proliferation signaling.

Phenanthriplatin also upregulated SNHG14 (small nucleolar RNA host gene 14) and to a greater degree than cisplatin (Table 1 and Table 5). SNHG14 is a long noncoding RNA which in A549 cells suppresses the microRNA, miR-34a, a negative regulator of high mobility group box 1 (HMGB1) mRNA, whose increased expression can promote anti-tumor drug resistance [24]. As higher expression of SNHG14 is associated with increased NSCLC cell proliferation, invasion, and migration, and silencing of SNHG14 causes sensitivity to cisplatin [24], our data suggests that the monofunctional complex could signal through the miR-34a-HMGB1 axis to promote resistance mechanisms if miR-34a expression occurs. 

We also found that the monofunctional complex downregulated two genes, DAPK1 (death associated protein kinase 1) and PFKFB3 (6-phosphofructo-2-kinase/fructose-2,6-biphosphatase 3), more than cisplatin in A549 cells (Table 1 and Table 5). Upregulation of DAPK1, a modulator of cell death in A549 cells, is associated with decreased cisplatin sensitivity [25]. As phenanthriplatin downregulates DAPK1 more than cisplatin (Table 5), this suggests that the monofunctional complex could signal via this kinase to enhance anticancer efficacy. PFKFB3 is an enzyme that regulates glycolysis and is upregulated in several cancers [26]. In A549 cells, inhibition of PFKFB3 sensitizes these cells to cisplatin treatment [26], indicating that as phenanthriplatin downregulates this phosphatase more than the bifunctional complex (Table 5), that it may have greater anticancer efficacy by modulating glycolytic signaling via preventing the activity of PFKFB3.

In both A549 and IMR90 cells, phenanthriplatin upregulated TNFRSF10D (tumor necrosis factor–related apoptosis-inducing ligand receptor superfamily member 10d) (Table 1, Figure 7), and this TNF-related apoptosis-inducing ligand (TRAIL) receptor gene is a downstream target of p53 signaling in NSCLC and breast cancer cell lines where its upregulation confers resistance to DNA-damaging agents [27]. As cisplatin can also signal through p53 to modulate cancer cell apoptosis [28], it is uncertain whether either platinum compound would confer a functional advantage in NSCLC treatment by acting through TRAIL receptor signaling.

Unlike cisplatin, phenanthriplatin upregulated ATP5MD (ATP synthase membrane subunit DAPIT), a gene that codes for a component of the mitochondrial H+-ATP synthase involved with mitochondrial oxidative phosphorylation, and with oncogenic function associated with elevated aerobic metabolism and epithelial-to-mesenchymal transition (EMT) [29] (Table 1, Figure 7). Increased ATP5MD activity has been reported to cause a shift to glycolysis in cancer cells and to promote EMT by alteration of Wnt/β catenin signaling where E-cadherin is replaced by N-cadherin [29,30]. Decreasing Wnt/β catenin signaling in A549 cells treated with cisplatin causes reduced migration and promotes apoptosis [31]. Thus, both complexes may integrate Wnt/β catenin using different gene pathways with uncertain comparative functional outcomes.

We also found that phenanthriplatin, but not cisplatin, downregulates two genes, AGAP1 (ArfGAP with GTPase domain, ankyrin repeat and PH domain 1) and DIAPH2 (diaphanous related formin 2), involved in cancer cell cytoskeletal remodeling (Table 4, Figure 7). AGAP1 binds to FilGAP, a Rac-specific GTPase-activating protein, facilitating its targeting, and decreased AGAP1 expression is associated with promotion of cell migration in breast cancer cells [32]. In colorectal cancer, DIAPH2 expression stimulates actin nucleation and microtubule stabilization, potentially controlling the cell cycle in a CDC42-independent manner [33]. Interestingly, studies conducted in A549 cells have shown that when CDC42 is upregulated, it causes increased proliferation and that CDC42 is directly regulated by the microRNA, miR-25, and that reduction of CDC42 is associated with enhanced cisplatin sensitivity [34]. Therefore, phenanthriplatin may act to modulate the cytoskeleton and cancer invasion through distinct pathways from those used by cisplatin, including a pathway incorporating AGAP1 and FilGAP signaling, while cisplatin signaling may not integrate this mechanism. However, phenanthriplatin may also act to prevent cancer cell cytoskeletal stability by downregulating DIAPH2, while cisplatin regulation of cancer cell microtubule stability may act instead through microRNA and CDC42 signaling to modulate A549 cancer cell proliferation.

Phenanthriplatin and cisplatin may modulate pathways integrating growth differentiation factor 15 (GDF15) signaling differently in A549 cells. Analysis of the NGS and ddPCR data set showed that phenanthriplatin caused a nonsignificant decrease in GDF15 expression in A549 cells; whereas cisplatin treatment was associated with a significant increase (Table 2, Table 3 and Table 4, Figure 7). Increased expression of the complement molecule, C5a, in NSCLC patients has been shown to activate signaling through its receptor, C5aR, and is associated with increased KLF5, GCN5, and GDF15 levels and promotion of A549 proliferation [35]. KLF5 and GCN5 encode proteins that form a complex which increases GDF15 gene transcription, a gene that acts as an oncogene to promote cell proliferation [35,36]. Our results could suggest that the monofunctional complex might act differently than cisplatin on components of the GDF15 signaling pathway to potentially prevent NSCLC proliferation; however, the precise target of the both platinum complexes on this signaling pathway are uncertain.

The monofunctional complex increased expression of the tumor suppressor, trefoil factor 1 (TFF1), more than cisplatin treatment did in A549 cells (Table 1 and Table 5, Figure 7). In gastrointestinal mucosa, TFF1 functions in protection and repair, while it typically has reduced expression in gastric cancer cell lines where it is regulated by DNA methylation and associated with increased p53 expression [37]. Furthermore, restoration of TFF1 in gastric cancer cells has been shown to activate p53 by downregulating miR-504, a negative regulator of p53 [38]. This result suggests that phenanthriplatin treatment may function upstream of p53 to promote cancer apoptosis. Interestingly, the ddPCR work showed that cisplatin, but not phenanthriplatin, increased TFF1 expression in IMR90 cells (Figure 7). Thus cisplatin, but not phenanthriplatin, might induce cell death in normal lung fibroblast cells, while the monofunctional complex could act as a more potent chemotherapeutic agent in some cancers.

We also identified genes that the two platinum compounds regulated differently. For example, phenanthriplatin treatment decreased the expression of the proto-oncogene, mouse double minute 2 homolog (MDM2), which in A549 and IMR90 cells, was increased by cisplatin treatment (Table 2, Table 4 and Table 5). Furthermore, the ddPCR results show that phenanthriplatin reduced G-protein-coupled receptor class C group 5 member A (GPRC5A) expression in A549 cells, but not in IMR90 cells, where cisplatin increased GPRC5A gene expression (Figure 7). As increased MDM2 expression suppresses GPRC5A, a tumor suppressor gene in A549 cells [39], our data might suggest that phenanthriplatin treatment could reduce MDM2 and thereby nullify the effect of GPRC5A to suppress tumor progression. GPRC5A encodes an endoplasmic reticulum localized protein that can reduce epidermal growth factor receptor (EGFR) signaling, and this receptor acts upstream of MDM2, which is a negative regulator of p53 [39,40]. These results suggest that phenanthriplatin may modulate distinct functional effects more than cisplatin by acting on both the MDM2 and GPRC5A components of a signal transduction loop that integrates p53 signaling. Phenanthriplatin might act on GPRC5A downstream of MDM2 to decrease EGFR function preventing MDM2’s role to negate p53, allowing p53 to act as a tumor suppressor in NSCLC. The GPRC5A ddPCR data (Figure 7) also suggests that phenanthriplatin may not prevent cell proliferation in normal lung cells; whereas cisplatin might impair this function in normal lung cells, but this conclusion is uncertain.

The current NGS and ddPCR data indicate that phenanthriplatin downregulated the gene thrombospondin type-1 domain-containing protein 1 (THSD1) in A549 cells, but, interestingly, cisplatin upregulated expression of this gene in the NSCLC cell line (Table 4, Figure 7). Downregulation of THSD1 is correlated with its methylation in colorectal cancer cell lines, and the gene encodes a transmembrane molecule thought to be involved in cell adhesion and angiogenesis with its loss being associated with metastatic tumor spread in breast cancer [41]. THSD1 is downregulated in A549 cells and other cancer cell lines, and may function as a tumor suppressor, but its role in cancer and interaction with other genes is not well understood [42]. However, THSD1 has recently been shown to form a protein complex with focal adhesion kinase (FAK), talin, and vinculin, where its role is to promote talin binding to FAK and functions in normal cell adhesion formation and attachment [43]. Therefore, our data suggests that unlike cisplatin, phenanthriplatin treatment might act to prevent the action of THSD1 in some cancers eliminating its role as a tumor suppressor, but the reason for the distinct action of the monofunctional complex compared to cisplatin on THSD1 is not clear.

In conclusion, we found that cisplatin and phenanthriplatin modulate a diverse set of genes that potentially modulate mechanisms associated with a wide variety of biological processes in both A549 non-small cell lung cancer and IMR90 non-cancerous lung fibroblast cell lines. Both the bifunctional and monofunctional complex may act through the same genes and potentially might cause similar cellular effects. However, we also found that phenanthriplatin regulates some genes whose expression is not altered by cisplatin treatment, and that the monofunctional complex can also modulate some genes differently than its bifunctional counterpart in both NSCLC and normal lung cells. The ability of phenanthriplatin to target unique genes and their associated mechanisms could possibly allow it to provide a clinical advantage in the treatment of cancers with appropriately matched genetic profiles.

## Figures and Tables

**Figure 1 cells-09-02637-f001:**
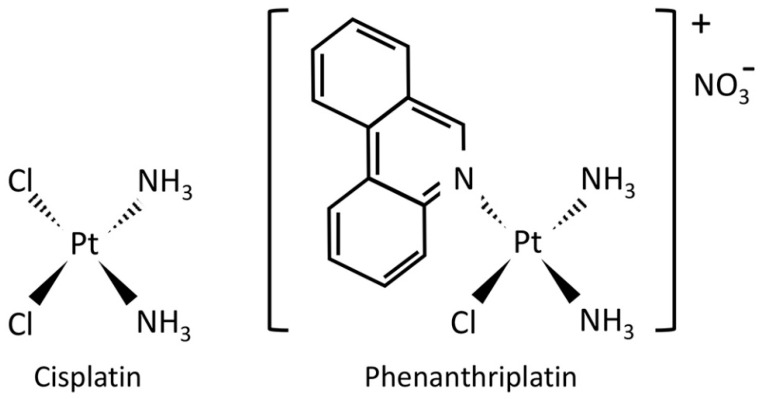
Chemical structures of cisplatin and phenanthriplatin. Cisplatin is a bifunctional platinum(II) complex whose anticancer activity is primarily related to its ability to intercalate and distort DNA leading to activation of DNA repair and cell death mechanisms. Phenanthriplatin is a heterocyclic-ligated monofunctional platinum(II) complex that is proposed to cause cancer cell death via blockage of gene transcription while evading DNA repair mechanisms.

**Figure 2 cells-09-02637-f002:**
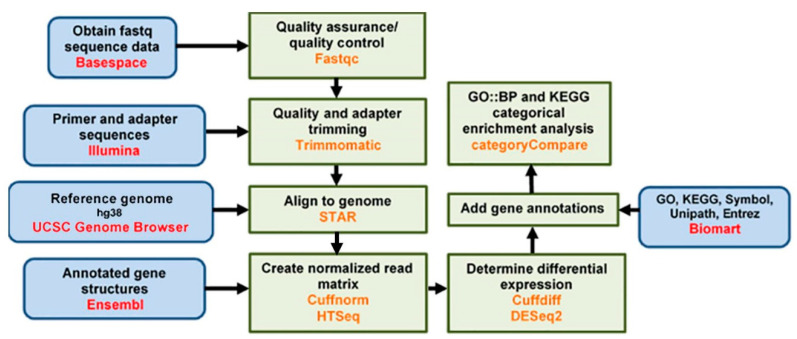
Next generation sequencing data analysis flow chart.

**Figure 3 cells-09-02637-f003:**
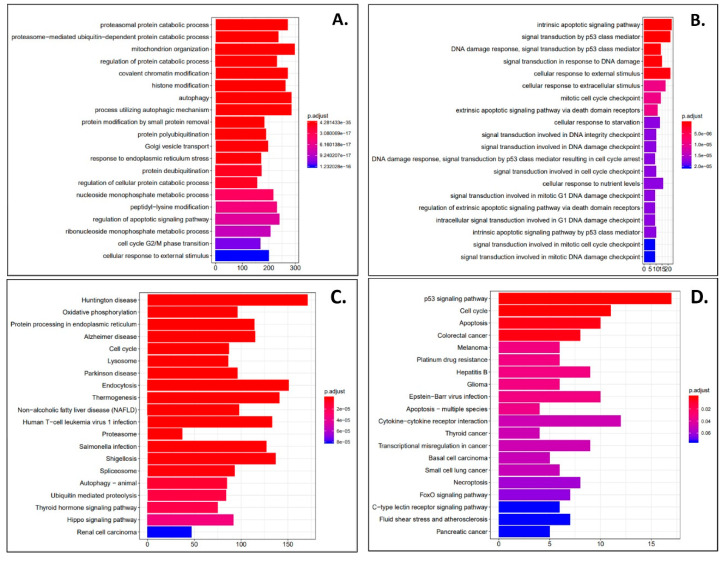
Phenanthriplatin and cisplatin modulate cellular pathways in A549 cells. (**A**) The top 20 enriched GO biological processes for the A549 phenanthriplatin versus A549 control comparison. (**B**) The top 20 enriched GO biological processes for the A549 cisplatin versus A549 control comparison. (**C**) The top 20 enriched KEGG pathways for the A549 phenanthriplatin versus A549 control comparison. (**D**) The top 20 enriched KEGG pathways for the A549 cisplatin versus A549 control comparison. The *x*-axes show the number of DEGs per GO biological process. Statistical analysis for KEGG pathway analysis is provided in the panels.

**Figure 4 cells-09-02637-f004:**
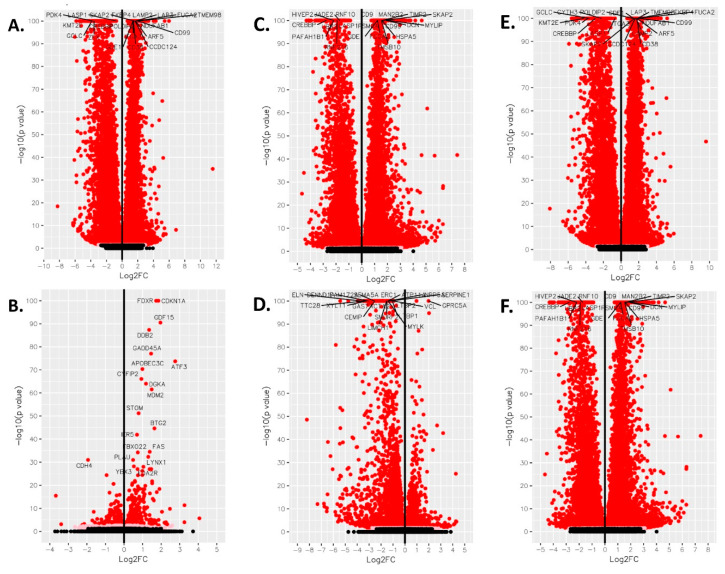
Volcano plot analysis of phenanthriplatin and cisplatin modulation of cellular pathways in A549, IMR90, and between A549 and IMR90 cells. Volcano plot for (**A**) A549 phenanthriplatin vs. A549 control, (**B**) A549 cisplatin vs. A549 control, (**C**) IMR90 phenanthriplatin vs. IMR90 control, (**D**) IMR90 cisplatin vs. IMR90 control, (**E**) A549 phenanthriplatin versus A549 cisplatin, and (**F**) IMR90 phenanthriplatin versus IMR90 cisplatin treatments. Log2 fold change on the x-axis is plotted against –log10(*p*-value) on the *y*-axis. Key: filled black circle = not significant, filled pink circle = *p* value < 0.05, filled red circle = *q* value < 0.05.

**Figure 5 cells-09-02637-f005:**
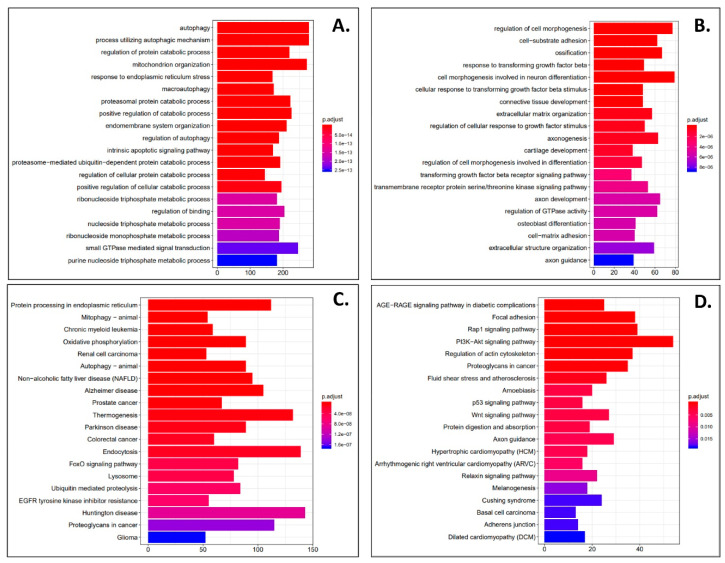
Phenanthriplatin and cisplatin modulate cellular pathways in IMR90 cells. (**A**) The top 20 enriched GO biological processes for the IMR90 phenanthriplatin versus IMR90 control comparison. (**B**) The top 20 enriched GO biological processes for the IMR90 cisplatin versus IMR90 control comparison. (**C**) The top 20 enriched KEGG pathways for the IMR90 phenanthriplatin versus IMR90 control comparison. (**D**) The top 20 enriched KEGG pathways for the IMR90 cisplatin versus IMR90 control comparison. The *x*-axes show the number of DEGs per GO biological process. Statistical analysis for KEGG pathway analysis is provided in the panels.

**Figure 6 cells-09-02637-f006:**
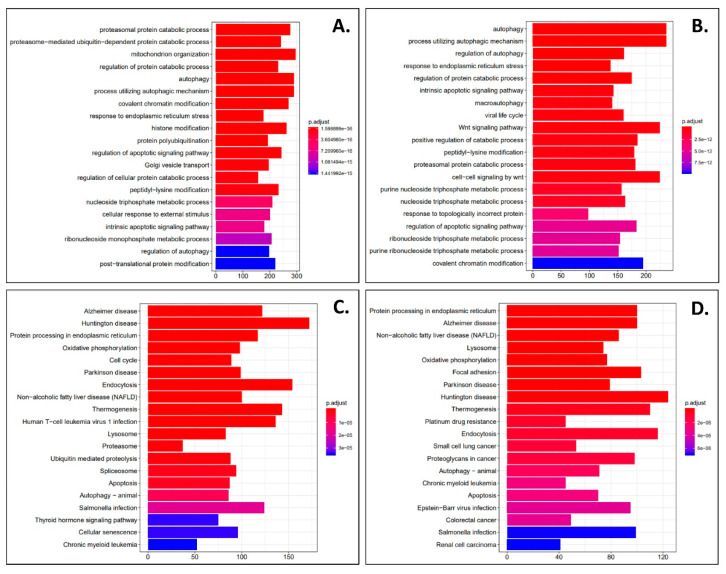
Comparative analysis of cellular pathway modulation between phenanthriplatin and cisplatin treatments in A549 and IMR90 cells. (**A**) The top 20 enriched GO biological processes for the A549 phenanthriplatin versus A549 cisplatin comparison. (**B**) The top 20 enriched GO biological processes for the IMR90 phenanthriplatin versus IMR90 cisplatin comparison. (**C**) The top 20 enriched KEGG pathways for the A549 phenanthriplatin versus A549 cisplatin comparison. (**D**) The top 20 enriched KEGG pathways for the IMR90 phenanthriplatin versus IMR90 cisplatin comparison. The *x*-axes show the number of DEGs per GO biological process. Statistical analysis for KEGG pathway analysis is provided in the panels.

**Figure 7 cells-09-02637-f007:**
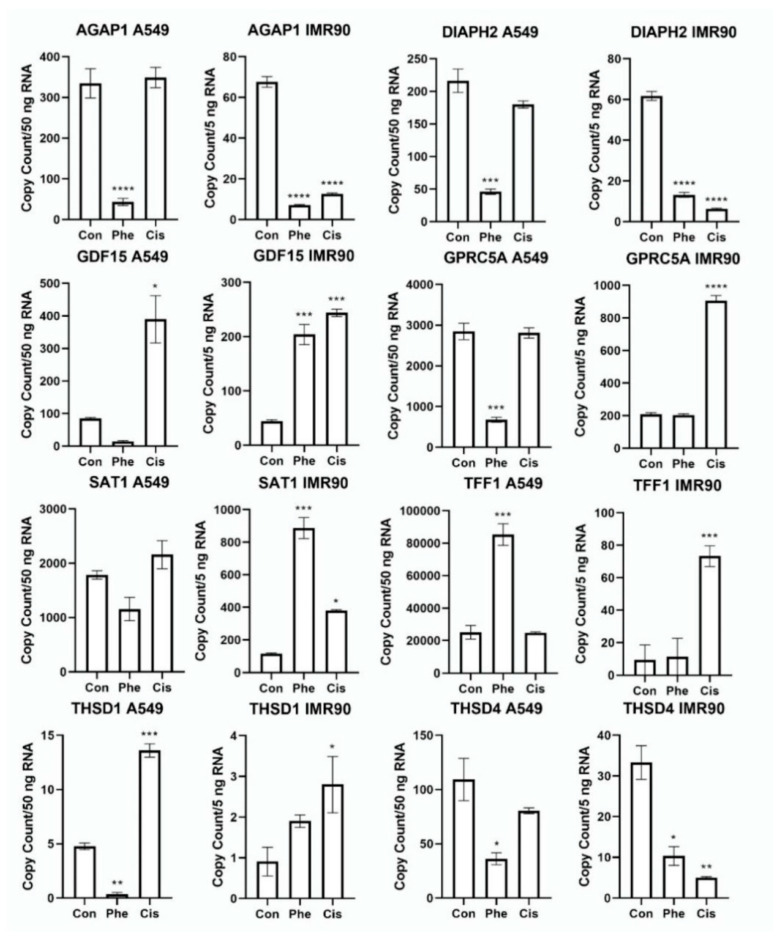
Validation of select regulated genes identified by next generation sequencing using droplet digital polymerase chain reaction in A549 and IMR90 cells. Mean expression of the genes, AGAP1, DIAPH2, GDF15, GPRC5A, SAT1, TFF1, THSD1, THSD4, as measured by ddPCR. Graph labeling: Con = control; Phe = phenanthriplatin; Cis = cisplatin. Y-axis represents copy count for 50 ng RNA sample. Mean (± SEM); N = 3; *p* < 0.05; “*” = *p* < 0.05; “**” = *p* < 0.01; “***” = *p* < 0.001; “****” = *p* < 0.0001.

**Table 1 cells-09-02637-t001:** Phenanthriplatin and cisplatin modulate gene regulation in A549 cells. Top 10 up- and downregulated DEGs for A549 phenanthriplatin versus A549 control. DEG data analysis: (*p* ≤ 0.05; *q* ≤ 0.05; log_2_FC ≥ 0; FPKM ≥ 1 in ≥ 3 samples; average FPKM ≥ 1 minimum count: 10).

Gene (↑)	Log_2_FC	*p*-Value	*q*-Value	Gene (↓)	Log_2_FC	*p*-Value	*q*-Value
ATF3	5.986	≤0.0001	≤0.0001	PFKFB3	−4.236	≤0.0001	≤0.0001
TFF1	4.067	≤0.0001	≤0.0001	LPCAT1	−4.240	≤0.0001	≤0.0001
RBP4	3.324	≤0.0001	≤0.0001	DAPK1	−4.305	≤0.0001	≤0.0001
S100A6	2.806	≤0.0001	≤0.0001	RAI1	−4.330	≤0.0001	≤0.0001
TNFRSF10D	2.760	≤0.0001	≤0.0001	ZNF609	−4.484	≤0.0001	≤0.0001
C12orf75	2.695	≤0.0001	≤0.0001	PDE4D	−4.590	≤0.0001	≤0.0001
TFPI2	2.691	≤0.0001	≤0.0001	B4GALT5	−4.665	≤0.0001	≤0.0001
ATP5MD	2.678	≤0.0001	≤0.0001	SIN3A	−4.809	≤0.0001	≤0.0001
SNHG14	2.667	≤0.0001	≤0.0001	HMOX1	−4.968	≤0.0001	≤0.0001
S100A4	2.659	≤0.0001	≤0.0001	TNFRSF1A	−5.852	≤0.0001	≤0.0001

**Table 2 cells-09-02637-t002:** Phenanthriplatin and cisplatin modulate gene regulation in A549 cells. Top 10 up- and downregulated DEGs for A549 cisplatin versus A549 control. DEG data analysis: (*p* ≤ 0.05; *q* ≤ 0.05; log_2_FC ≥ 0; FPKM ≥ 1 in ≥ 3 samples; average FPKM ≥ 1 minimum count: 10).

Gene (↑)	Log_2_FC	*p*-Value	*q*-Value	Gene (↓)	Log_2_FC	*p*-Value	*q*-Value
ATF3	2.757	1.880 × 10^−74^	4.441 × 10^−71^	MCM5	−0.482	7.133 × 10^−11^	9.723 × 10^−9^
GDF15	1.962	2.950 × 10^−91^	1.393 × 10^−87^	SLIT3	−0.530	6.219 × 10^−13^	1.101 × 10^−10^
CDKN1A	1.852	2.829 × 10^−142^	4.010 × 10^−138^	TRAPPC9	−0.565	2.361 × 10^−21^	1.014 × 10^−18^
FDXR	1.728	2.276 × 10^−111^	1.613 × 10^−107^	SAPCD2	−0.686	1.617 × 10^−12^	2.730 × 10^−10^
BTG2	1.639	2.434 × 10^−45^	2.875 × 10^−42^	KCNMA1	−0.710	2.770 × 10^−12^	4.620 × 10^−10^
MDM2	1.490	2.982 × 10^−62^	4.227 × 10^−59^	MSRA	−0.769	4.236 × 10^−10^	5.177 × 10^−8^
SESN2	1.467	8.864 × 10^−28^	5.815 × 10^−25^	MCM6	−0.782	1.516 × 10^−17^	4.478 × 10^−15^
GADD45A	1.458	8.776 × 10^−78^	2.488 × 10^−74^	GPC6	−0.940	3.773 × 10^−25^	2.057 × 10^−22^
FAS	1.384	3.439 × 10^−35^	3.482 × 10^−32^	CDH4	−1.940	1.156 × 10^−31^	9.432 × 10^−29^
DDB2	1.350	4.346 × 10^−88^	1.540 × 10^−84^	LSAMP	−3.677	3.272 × 10^−16^	8.434 × 10^−14^

**Table 3 cells-09-02637-t003:** Phenanthriplatin and cisplatin modulate gene regulation in IMR90 cells. Top 10 up- and downregulated DEGs for IMR90 phenanthriplatin versus IMR90 control. DEG data analysis: (*p* ≤ 0.05; *q* ≤ 0.05; log_2_FC ≥ 0; FPKM ≥ 1 in ≥ 3 samples; average FPKM ≥ 1 minimum count: 10).

Gene (↑)	Log_2_FC	*p*-Value	*q*-Value	Gene (↓)	Log_2_FC	*p*-Value	*q*-Value
ATF3	5.364	≤0.0001	≤0.0001	HMGA2	−3.439	≤0.0001	≤0.0001
SAT1	4.094	≤0.0001	≤0.0001	ASAP1	−3.543	≤0.0001	≤0.0001
GDF15	3.657	≤0.0001	≤0.0001	GSK3B	−3.549	≤0.0001	≤0.0001
GADD45A	3.433	≤0.0001	≤0.0001	LBH	−3.571	≤0.0001	≤0.0001
NPTX1	2.941	7.459 × 10^−307^	2.830 × 10^−304^	NRG1	−3.680	≤0.0001	≤0.0001
PCNA	2.344	≤0.0001	≤0.0001	IGF2BP2	−4.246	≤0.0001	≤0.0001
PEG10	2.312	≤0.0001	≤0.0001	COL1A1	−4.330	≤0.0001	≤0.0001
TM4SF1	2.103	≤0.0001	≤0.0001	SLC38A2	−4.409	≤0.0001	≤0.0001
TFPI2	2.007	≤0.0001	≤0.0001	ZMIZ1	−4.422	≤0.0001	≤0.0001
TNFRSF10D	2.006	≤0.0001	≤0.0001	SMURF2	−4.866	≤0.0001	≤0.0001

**Table 4 cells-09-02637-t004:** Phenanthriplatin and cisplatin modulate gene regulation in IMR90 cells. Top 10 up- and downregulated DEGs for IMR90 cisplatin versus IMR90 control. DEG data analysis: (*p* ≤ 0.05; *q* ≤ 0.05; log_2_FC ≥ 0; FPKM ≥ 1 in ≥ 3 samples; average FPKM ≥ 1 minimum count: 10).

Gene (↑)	Log_2_FC	*p*-Value	*q*-Value	Gene (↓)	Log_2_FC	*p*-Value	*q*-Value
THSD1	2.690	9.073 × 10^−47^	5.963 × 10^−45^	PLXDC2	−2.875	7.181 × 10^−249^	9.512 × 10^−246^
GDF15	2.016	1.914 × 10^−95^	3.330 × 10^−93^	TRAPPC9	−2.988	3.254 × 10^−240^	3.503 × 10^−237^
GPRC5A	1.975	5.924 × 10^−103^	1.214 × 10^−100^	PRKCA	−3.083	3.001 × 10^−224^	2.720 × 10^−221^
CYP1B1	1.556	2.229 × 10^−44^	1.396 × 10^−42^	EXOC4	−3.317	9.567 × 10^−293^	2.353 × 10^−289^
PLCXD1	1.553	1.755 × 10^−51^	1.244 × 10^−49^	PTPRG	−3.364	≤0.0001	≤0.0001
SAT1	1.444	1.116 × 10^−79^	1.467 × 10^−77^	LRBA	−3.455	4.344 × 10^−275^	8.312 × 10^−272^
FDXR	1.417	8.633 × 10^−58^	7.182 × 10^−56^	NAALADL2	−3.479	3.697 × 10^−212^	3.184 × 10^−209^
MDM2	1.246	4.780 × 10^−44^	2.919 × 10^−42^	THSD4	−3.693	≤0.0001	≤0.0001
CDKN1A	1.212	1.385 × 10^−63^	1.332 × 10^−61^	AGAP1	−3.956	≤0.0001	≤0.0001
PLK2	1.176	1.313 × 10^−46^	8.603 × 10^−45^	DIAPH2	−4.744	3.869 × 10^−244^	4.760 × 10^−241^

**Table 5 cells-09-02637-t005:** Comparative analysis of gene regulation between phenanthriplatin and cisplatin treatments in A549 and IMR90 cells. Top 10 up-and downregulated DEGs for A549 phenanthriplatin versus A549 cisplatin. DEG data analysis: (*p* ≤ 0.05; *q* ≤ 0.05; log_2_FC ≥ 0; FPKM ≥ 1 in ≥ 3 samples; average FPKM ≥ 1 minimum count: 10).

Gene (↑)	Log_2_FC	*p*-Value	*q*-Value	Gene (↓)	Log_2_FC	*p*-Value	*q*-Value
TFF1	4.503	≤0.0001	≤0.0001	PFKFB3	−4.165	≤0.0001	≤0.0001
RBP4	3.360	≤0.0001	≤0.0001	DAPK1	−4.210	≤0.0001	≤0.0001
CST1	2.984	≤0.0001	≤0.0001	NR3C1	−4.246	≤0.0001	≤0.0001
SNHG14	2.741	≤0.0001	≤0.0001	RAI1	−4.316	≤0.0001	≤0.0001
TFPI2	2.740	≤0.0001	≤0.0001	PDE4D	−4.425	≤0.0001	≤0.0001
SERF2	2.733	≤0.0001	≤0.0001	ZNF609	−4.451	≤0.0001	≤0.0001
S100A6	2.722	≤0.0001	≤0.0001	MDM2	−4.538	≤0.0001	≤0.0001
S100A4	2.675	≤0.0001	≤0.0001	SIN3A	−4.629	≤0.0001	≤0.0001
FSTL3	2.661	≤0.0001	≤0.0001	B4GALT5	−4.724	≤0.0001	≤0.0001
COX8A	2.654	≤0.0001	≤0.0001	TNFRSF1A	−5.596	≤0.0001	≤0.0001

**Table 6 cells-09-02637-t006:** Comparative analysis of gene regulation between phenanthriplatin and cisplatin treatments in A549 and IMR90 cells. Top 10 up-and downregulated DEGs for IMR90 phenanthriplatin versus IMR90 cisplatin. DEG data analysis: (*p* ≤ 0.05; *q* ≤ 0.05; log_2_FC ≥ 0; FPKM ≥ 1 in ≥ 3 samples; average FPKM ≥ 1 minimum count: 10).

Gene (↑)	Log_2_FC	*p*-Value	*q*-Value	Gene (↓)	Log_2_FC	*p*-Value	*q*-Value
THSD4	3.658	≤0.0001	≤0.0001	PUM1	−2.792	≤0.0001	≤0.0001
NPTX1	3.163	≤0.0001	≤0.0001	TRAM2	−2.797	≤0.0001	≤0.0001
P4HA3	2.809	≤0.0001	≤0.0001	CDC42EP3	−2.833	≤0.0001	≤0.0001
CEMIP	2.668	≤0.0001	≤0.0001	PSME4	−2.860	≤0.0001	≤0.0001
GADD45A	2.616	≤0.0001	≤0.0001	FOXP1	−3.229	2.013 × 10^−283^	8.117 × 10^−281^
SERPINE2	2.045	≤0.0001	≤0.0001	GSK3B	−3.275	4.903 × 10^−290^	2.314 × 10^−287^
DCN	1.906	≤0.0001	≤0.0001	SMURF2	−3.359	≤0.0001	≤0.0001
COL4A2	1.875	≤0.0001	≤0.0001	NRG1	−3.561	≤0.0001	≤0.0001
TFPI2	1.692	≤0.0001	≤0.0001	IGF2BP2	−3.569	4.865 × 10^−272^	1.711 × 10^−269^
PEG10	1.689	≤0.0001	≤0.0001	SLC38A2	−4.173	≤0.0001	≤0.0001

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
