# Peer review of "RNA-Seq Analysis of Cisplatin and the Monofunctional Platinum(II) Complex, Phenanthriplatin, in A549 Non-Small Cell Lung Cancer and IMR90 Lung Fibroblast Cell Lines"

_cells, 2020, doi:10.3390/cells9122637_

Round 1

Reviewer 1 Report

The manuscript by Monroe et al describes data acquired by analyzing gene expression profiles in two cisplatin and phenanthriplatin treated cell lines using RNA-seq.

Comments:

  • This is a relatively small experiment presented in a very extensive way. Several parts of the manuscript could be shortened. It is not necessary to describe the full procedure of RNA extraction, library preparation, sequencing and ddPCR if the authors just followed the vendors' recommendations. There is also a lot of redundancy in the results; figure presented data are also included in the text.
  • It would be useful to analyze more cell lines to see how robust and general the findings are.
  • The authors say in the introduction that mRNA was extracted from the cell lines. Wasn't it rather total RNA? The mRNA was fished out with oligo (dT) beads during the library prep if I'm correct.
  • How was the concentration of 5uM chosen for the cisplatin and phenanthriplatin treatments. Were titration experiments completed upfront?
  • Section 2.4. first sentence: 'The pooled library was prepared and sequenced' would be better.
  • Was the control normalized data used for the comparison of different treatments in the same cell line?
  • Figure 2: there is a disconnectivity in the map (missing arrow?). Multiple different reference genomes are displayed while the text only refers to hg38.
  • Figures 3, 4 and 5 contain unreadable text.
  • References are not numbered.

Author Response

Reviewer 1

Comments and Suggestions for Authors

The manuscript by Monroe et al describes data acquired by analyzing gene expression profiles in two cisplatin and phenanthriplatin treated cell lines using RNA-seq.

Comments:

This is a relatively small experiment presented in a very extensive way. Several parts of the manuscript could be shortened. It is not necessary to describe the full procedure of RNA extraction, library preparation, sequencing and ddPCR if the authors just followed the vendors' recommendations. There is also a lot of redundancy in the results; figure presented data are also included in the text. 

Our response: We greatly shortened the RNA extraction and library preparation sections and combined them into one section, we shortened the length of the RNA sequencing and ddPCR sections.  Also, we greatly reduced redundant material from the figures that was stated in the results section [lines 124-181; 190-193; 201-202; 207-208; 257-276; 324-336; 362-378]. 

It would be useful to analyze more cell lines to see how robust and general the findings are. 

Our response:  Although we agree that more cell lines would provide a broader basis for analysis, our initial objective was to gather preliminary data in relevant cancerous and non-cancerous cell lines, and pursuant to this aim we adopted a project methodology with two carefully selected cell lines which allowed us to test the effects of the platinum compounds in both non-small cell lung cancer and a non-cancer lung fibroblast cell line.  We believe that this experimental methodology provides a significant and successful basis of comparison to assess the gene regulatory effects of cisplatin, phenanthriplatin and control treatments in both lung cancer and normal lung cells.    

The authors say in the introduction that mRNA was extracted from the cell lines. Wasn't it rather total RNA? The mRNA was fished out with oligo (dT) beads during the library prep if I'm correct. 

Our response: We corrected the statement in the introduction to say that the extracted RNA was total RNA [Line 86].

How was the concentration of 5uM chosen for the cisplatin and phenanthriplatin treatments. Were titration experiments completed upfront? 

Our response: This concentration has been used by us and others to determine the cellular and nuclear uptake of these compounds in these and other cell lines (Park et al., Phenanthriplatin, a monofunctional DNA-binding platinum anticancer drug candidate with unusual potency and cellular activity profile, Proceedings of the National Academy of Sciences, 2012, 109(30):11987-92; Monroe et al., Anti-cancer characteristics and ototoxicity of platinum(II) amine complexes with only one leaving ligand, PLoS One 2018 13(3):e0192505.2018).  We added this information to the Materials and Methods section with the references [Lines 119-120].

Section 2.4. first sentence: 'The pooled library was prepared and sequenced' would be better. 

Our response: We introduced the reviewer’s text recommendation [Line 185].

Was the control normalized data used for the comparison of different treatments in the same cell line? 

Our response: We used DESeq2 on the raw counts for normalization, which creates a scaling factor based on median gene expression across the samples.  This is a standard method for normalization, which avoids problems associated with normalizing to a control.  We noted this in the Data Analysis section of the Materials and Methods and introduced a reference (Anders and Huber, 2010) regarding this form of analysis there [Lines 233-234].

Figure 2: there is a disconnectivity in the map (missing arrow?). Multiple different reference genomes are displayed while the text only refers to hg38. 

Our response: We added an arrow to provide complete connectivity in the flow chart and also corrected the figure to show that only reference genome hg38 was used [Lines 103-104].

Figures 3, 4 and 5 contain unreadable text. 

Our response: We broke up figures 3, 4, and 5 into tables and figures with much larger font that should now be readable.  Also, table and figure numbering has been adjusted in the manuscript [Lines 282-301; 302-310; 339-360; 384-407].

References are not numbered. 

Our response:  We introduced reference numbering compatible with journal guidelines [Lines 606-706].

Reviewer 2 Report

I am afraid, I am not supporting publishing of this manuscript. It appears immature and unfinished to me. This is a very descriptive study that compares the transcriptome profiles of cisplatin and phenanthriplatin in a very simplistic way. There is hardly any analysis of the results. Hence, the authors present a dataset rather than a scientific study.

There should be plenty of data available on the effects of cisplatin on the cellular transcriptome, probably including data on the effect of cisplatin on A549 cells (and perhaps also IMR90). Hence, the data should be compared to those by others to put them into context.

The authors use both cisplatin and phenanthriplatin at the same concentration of 5µM and perform RNA-Seq after 24h. However, it is not clear whether both drugs share the same molecular potency and the same activity kinetics. An in-depth characterisation of the action of both drugs in the cell line models would have been needed to put the results into context.

In the Methods section, the authors refer to four treatment modalities, but I can only see three? Cisplatin, phenanthriplatin, and control?

Author Response

Reviewer 2

Comments and Suggestions for Authors

I am afraid, I am not supporting publishing of this manuscript. It appears immature and unfinished to me. This is a very descriptive study that compares the transcriptome profiles of cisplatin and phenanthriplatin in a very simplistic way. There is hardly any analysis of the results. Hence, the authors present a dataset rather than a scientific study. 

Our response: We appreciate the reviewer’s concerns and have attached corrections which are described in bold below.

There should be plenty of data available on the effects of cisplatin on the cellular transcriptome, probably including data on the effect of cisplatin on A549 cells (and perhaps also IMR90). Hence, the data should be compared to those by others to put them into context. 

Our response:  We agree with the reviewer’s suggestion, and after performing an extensive literature search, we added a presentation of the effect of cisplatin on the transcriptome to the discussion which is focused on the effect of cisplatin on A549 cells.  We have further extended this discussion to integrate our next generation sequencing results and compare the effect of cisplatin with our A549 phenanthriplatin data and IMR90 cell line data [Lines 457-484].

The authors use both cisplatin and phenanthriplatin at the same concentration of 5µM and perform RNA-Seq after 24h. However, it is not clear whether both drugs share the same molecular potency and the same activity kinetics. An in-depth characterisation of the action of both drugs in the cell line models would have been needed to put the results into context. 

Our response: The 5 µM concentration has been used previously (Park et al., Phenanthriplatin, a monofunctional DNA-binding platinum anticancer drug candidate with unusual potency and cellular activity profile, Proceedings of the National Academy of Sciences, 2012, 109(30):11987-92; Monroe et al., Anti-cancer characteristics and ototoxicity of platinum(II) amine complexes with only one leaving ligand, PLoS One 2018 13(3):e0192505.2018) to determine the cellular and nuclear uptake of these complexes in these and other cell lines, and we have added this information to the Materials and Methods section with references.  As there is a body of work that already exists concerning cisplatin and phenanthriplatin cellular uptake and compartmental distribution, reaction kinetics with nucleotide targeting and the molecular potency in several cell lines, including the ones studied in this project, we added a synopsis of this information to the beginning of the discussion with references [Lines 119-120; 427-449].

In the Methods section, the authors refer to four treatment modalities, but I can only see three? Cisplatin, phenanthriplatin, and control?

Our response:  We corrected the number of treatment modalities to accurately reflect that there were only three and any statements affected by this error [Lines 219, 222].

Reviewer 3 Report

The manuscript describes the gene expression by cisplatin and phenanthriplatin in lung cancer cells and non-malignant lung cells. Possible functions and pathways were identified. Certain differences were observed for cells treated with cisplatin or with phenanthriplatin as well as between malignant and non-malignant cells. The results are meaningful and the provided discussion of the results is appropriate. I recommend acceptance after minor revision:

Figure 1: The structure of phenanthriplatin is incorrect. As mentioned in the main text, it is a cationic complex so please add the positive charge and the counterion/anion.

Figures 3-5: The texts in these figures are hard to read because the letters are so small. Please improve these figures.

Author Response

Reviewer 3

Comments and Suggestions for Authors

The manuscript describes the gene expression by cisplatin and phenanthriplatin in lung cancer cells and non-malignant lung cells. Possible functions and pathways were identified. Certain differences were observed for cells treated with cisplatin or with phenanthriplatin as well as between malignant and non-malignant cells. The results are meaningful and the provided discussion of the results is appropriate. I recommend acceptance after minor revision:

Figure 1: The structure of phenanthriplatin is incorrect. As mentioned in the main text, it is a cationic complex so please add the positive charge and the counterion/anion. 

Our response:  We have provided a new, higher resolution figure with the requested corrections [Lines 52-53].

Figures 3-5: The texts in these figures are hard to read because the letters are so small. Please improve these figures. 

Our response:  We broke up figures 3, 4, and 5 into tables and figures with much larger font that should now be readable.  Also, table and figure numbering has been adjusted in the manuscript (see also reviewer 1 comment for this point) [Lines 282-301; 302-310; 339-360; 384-407].

Round 2

Reviewer 1 Report

No further comment

Reviewer 2 Report

I am afraid, I do not understand how you can interpret the RNA-Seq data without knowing the impact of the drugs on cell viability. This is needed to put the findings into context.
